# Recite, Reconstruct, Recollect: Memorization in LMs as a Multifaceted Phenomenon

**USVSN Sai Prashanth**[*,1]    **Alvin Deng**[*,1,4]    **Kyle O'Brien**[*,1,2]    **Jyothir S V**[*,1,3]
**Mohammad Aflah Khan**[1,6,7]    **Jaydeep Borkar**[5]    **Christopher A. Choquette-Choo**[8]
**Jacob Ray Fuehne**[9]    **Stella Biderman**[1]    **Tracy Ke**[†,10]    **Katherine Lee**[†,8]    **Naomi Saphra**[†,10,11]
[1]EleutherAI    [2]Microsoft    [3]New York University    [4]DatologyAI    [5]Northeastern University
[6]MPI-SWS    [7]IIIT Delhi    [8]Google DeepMind    [9]University of Illinois at Urbana-Champaign
[10]Harvard University    [11]Kempner Institute
 **Correspondence:** katherinelee@google.com and nsaphra@fas.harvard.edu

## Abstract

Memorization in language models is typically treated as a homogenous phenomenon, neglecting the specifics of the memorized data. We instead model memorization as the effect of a set of complex factors that describe each sample and relate it to the model and corpus. To build intuition around these factors, we break memorization down into a taxonomy: recitation of highly duplicated sequences, reconstruction of inherently predictable sequences, and recollection of sequences that are neither. We demonstrate the usefulness of our taxonomy by using it to construct a predictive model for memorization. By analyzing dependencies and inspecting the weights of the predictive model, we find that different factors influence the likelihood of memorization differently depending on the taxonomic category.

## 1 Introduction

The existing literature on Language Model (LM) **memorization**[1]—the tendency to generate exact copies of training samples at test time—varies widely in stated motivation. Papers might focus on copyright (Shi et al., 2023; Karamolegkou et al., 2023; Meeus et al., 2024), privacy (Carlini et al., 2018; 2022b; Brown et al., 2022; Mireshghallah et al., 2022), or scientifically understanding how interpolation (Mallinar et al., 2022) leads to generalization (Feldman, 2021; Tirumala et al., 2022; Henighan et al., 2023a). Although these objectives share commonalities, they also drive distinct and sometimes contradictory notions of memorization. To disentangle these motivations and to articulate the factors that determine or signal memorization, we propose a taxonomy inspired by colloquial distinctions of memorization behavior in humans.

Our taxonomy, illustrated in Fig. 1, defines three types of LM memorization based on colloquial descriptions of human memorization. Humans **recite** direct quotes that they commit to memory through repeated exposure, so LMs recite highly duplicated sequences. Humans **reconstruct** a passage by remembering a general pattern and filling in the gaps, so LMs reconstruct inherently predictable boilerplate templates. Humans sporadically **recollect** an episodic memory or fragment after a single exposure, so LMs recollect other sequences seen rarely during training.

We use our taxonomy in a variety of experiments that highlight the multifaceted nature of memorization. In summary:

- We introduce an intuitive taxonomy and heuristics for categorizing memorized data.
- By comparing memorized and unmemorized distributions, we assess how a variety of corpus-wide statistics, datum-level metrics, and representational differences influence the

---

[*] Equal contribution
[†] Equal contribution
[1]As defined by www.genlaw.org/glossary.html.

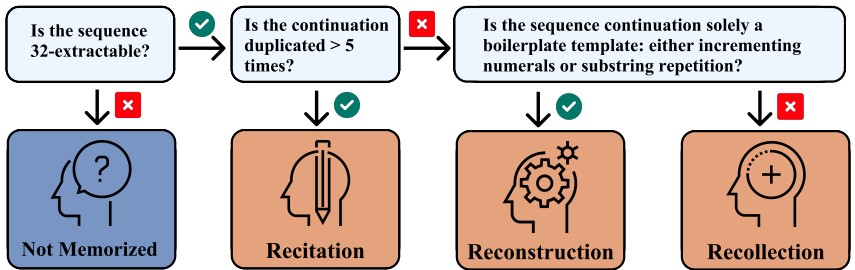

Figure 1: Our intuitive memorization taxonomy has three categories determined by simple heuristics.

likelihood of a given sequence being memorized. Our dependency tests confirm existing findings that low perplexity is strongly associated with memorization—though not equally for all memorized examples. This fact guides our heuristic for partitioning memorized data into a recitation category.

- We study scaling factors in memorization by monitoring each taxonomic category over the course of training and across model sizes. The number of memorized sequences increases with training time and model size, regardless of taxonomic category. Recollection, however, sees the fastest increase—and this outsize growth cannot be attributed solely to repeated exposures to rare sequences or to random memorization.

- To demonstrate the value of our taxonomy, we train logistic regressions to predict the likelihood of memorization for candidate sequences from each memorization category. This predictive model outperforms both a simple baseline with no taxonomy and a model that uses a taxonomy optimized by searching for the best set of mediating factors. These experiments show that the intuitions behind our taxonomy can improve on more generic approaches.

- We highlight differences between categories by exploring statistical dependencies, finding recitation is enabled by low-perplexity prompts and recollection is constrained by the presence of rare tokens.

## 2 EXPERIMENTS

In this section, we detail the definitions and data we use to analyze varying factors in memorization.

**Defining Memorization** There are multiple competing definitions for memorization (Zhang et al., 2021; Ippolito et al., 2022). Because our experiments employ memorization data released by Biderman et al. (2023a), we use their preferred definition of *k-extractable* memorization (Carlini et al., 2022a) with $k = 32$. A sample is $k$-extractable if the LM, when prompted with the first $k$ tokens, generates the following $k$ tokens verbatim.

**Language Models** We study memorization across model scale and training timing using the deduplicated Pythia models (Biderman et al., 2023b), which range in size from 70M to 12B parameters[2] trained on a deduped version of The Pile (Gao et al., 2020). Data order is fixed across runs, enabling causal claims about the effect of model scale on memorization.

**Datasets** Our **memorized sample** is a public list of sequences memorized by Pythia, released by Biderman et al. (2023a). Unlike other works that estimate whether a generation is from a model's training set using predictive techniques (Carlini et al., 2020; Shi et al., 2023; Yang et al., 2024), this dataset contains all 32-extractable samples from the Pile, verified by referencing the training data (Gao et al., 2020). We also collect a **representative sample** by taking a random 3% subset of The Pile, retaining the first 64 tokens of each sequence. Some analysis also considers an **unmemorized distribution** estimated by subtracting the memorized data distribution from the entire Pile, as inferred from the representative sample.

---

[2]Excluding the 160M parameter model, as its memorization dataset exhibits outlier behavior that could be either a buggy data artifact or a real phenomenon, but is regardless outside of the scope of our work.

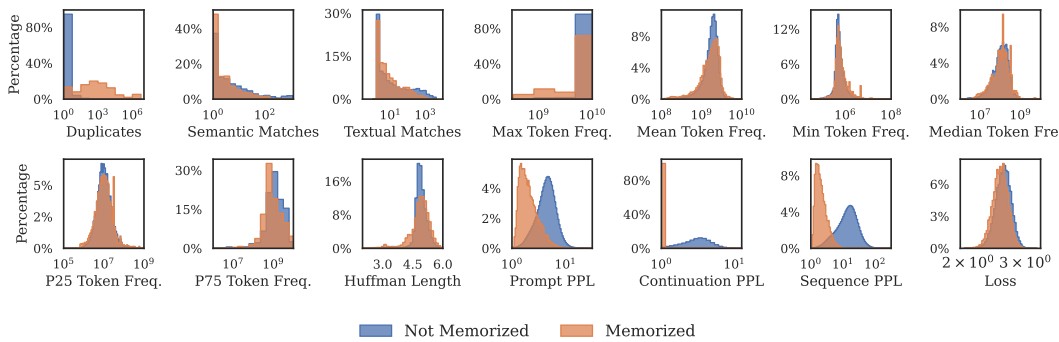

Figure 2: Histogram of various properties of interest (described in Section 3) for memorized and unmemorized (estimated by assuming the representative dataset's statistics hold for the Pile) samples.

## 3 POTENTIAL FACTORS IN MEMORIZATION

We consider a number of possible factors in whether a given sequence is memorized. These factors are based on corpus statistics, datum statistics intrinsic to that sample, or model perplexity. Features may be computed over the first 32 tokens (the **prompt**); the last 32 tokens (the **continuation**); and the **full sequence** of 64 tokens subsampled from the training data. Implementation details are provided in Appendix A.

Many of these properties have different distributions for memorized and unmemorized data. Fig. 2 illustrates these differences, highlighting that for some properties, the memorized distribution is more concentrated. Other properties—in particular perplexity and number of duplicates in the training corpus—have memorized and unmemorized distributions with visibly different medians. Where the distributions differ, the property in question is likely to influence memorization, an assumption which we employ predictively in Section 6.

### 3.1 CORPUS STATISTICS

Some factors relate a given sequence to the entire training corpus. Overall, the following features illustrate how memorization is influenced by various types of duplication.

**Duplicates** For each 32-token window in any 2049-token sequence seen during training, we count the number of duplicates in the Pile.

**Semantic Matches** To assess the prevalence of semantically similar samples in training, we generate document embeddings for each full sequence using SBERT and count the number of sequences with cosine similarity $\geq 0.8$, out of all 64-token sequences in The Pile. These sequences are semantically similar but may not be exact token-level duplicates.

**Textual Matches** We filter the set of semantic matches for a given target sequence to identify those with a low Levenshtein edit distance in their prompts (Levenshtein et al., 1966) from the target sequence. These matches flag slight variations on boilerplate prompts. We compute edit distance at the character level, thereby accounting for different tokenizations of identical sequences.

**Token frequency** We also compute summary statistics about the corpus-wide frequency of individual tokens in the sequence: mean, median, maximum, minimum, and 25th / 75th percentile counts.

### 3.2 SEQUENCE PROPERTIES

Because some sequences are inherently easier to encode, we also consider factors determined by intrinsic metrics on the sample itself.

**Templating** A sample is classified as templating if it follows a predictable pattern. We do not comprehensively consider all possible templates, but focus on two common patterns defined by handcrafted heuristics:

- **Repeating:** Consisting only of a short repeating sequence of tokens, e.g., "Go Go Go ...". Zhang et al. (2021) previously discussed repetitive templates as a common feature of apparently memorized data which was not classified as counterfactually memorized.

- **Incrementing:** Consisting of incrementing numerical sequences. For example, consider the sequence "23: 0xf1, 24: 0xf2, 25: 0xf3", a set of interspersed numerical sequences with repeating separators.

**Compressibility** We use Huffman Coding (Huffman, 1952) length to measure how easily a sequence is compressed. Compressibility generalizes repeating templates to cases where minor variations on repeating patterns must be memorized. The connection between learning, memorization, and compression is drawn from the existing literature: Carlini et al. (2020) attempts to filter out sequences that are "easy" to produce by comparing zlib compression with perplexity to identify memorized training data.

## 3.3 Perplexity

We compute average perplexity across tokens on the prompt, continuation, and full sequence. The importance of perplexity is one of the most reproduced results in memorization research (Zhang et al., 2021; Carlini et al., 2018) and we confirm that low perplexity sequences are far more likely to be memorized than high perplexity sequences (Fig. 2). Perplexity is the only factor we consider that relates to model behavior, rather than being intrinsic to the data.

## 4 Memorization Taxonomy

To analyze the fundamental causes of $k$-extracted memorization, we subdivide memorized samples into three types. The following rules categorize a sample as a **candidate** for recitation, reconstruction, or recollection; candidates memorized by the model are therefore respectively recited, reconstructed, or recollected.

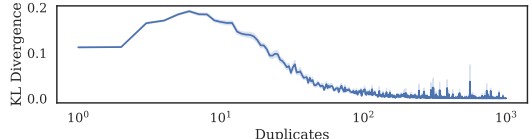

Figure 3: KL divergence between generation perplexity of memorized and non-memorized examples for Pythia 12B with bootstrapped confidence intervals. Non-memorized samples are treated as the reference distribution. Divergence is highest for sequences with 6 duplicates, while highly duplicated sequences have near-identical memorized and unmemorized distributions.

## 4.1 Recitation

The existing memorization literature agrees that the duplication of a sequence across the training corpus is strongly correlated with its memorization (Lee et al., 2021; Kandpal et al., 2022). For example, LMs produce verbatim copies of bible quotes or software licenses that are commonly duplicated.

We consider a sample to be a recitation candidate if it is highly duplicated in the training corpus. Model perplexity is a good predictor of memorization on rare sequences because the perplexity distributions are more different on memorized and unmemorized data with few duplicates (Fig. 3). For highly duplicated sequences, however, perplexity is no longer a good predictor of whether the sequence is memorized or not. We therefore define a recitation candidate as a sequence with at least 6 duplicates because the three-way relationship between perplexity, memorization, and duplicate count differs before and after that maximum divergence point. We test this threshold against others in Appendix G and find that it matches or beats examples of other small thresholds.

LMs memorize a wide variety of highly duplicated texts, as shown in the example of Appendix F. Recited natural language text largely comprises webpage boilerplate text, liturgy, and software licenses or other legalese. Table 3, which includes random samples of natural language recitation, includes all of these common cases. Recited code text, as seen in Appendix 4, is largely web development (HTML, CSS, JavaScript, etc.) boilerplate that describes common elements or derives from popular webpage templates.

## 4.2 RECONSTRUCTION

Are all perfectly reproduced sequences truly "memorized"? We consider cases that may be spuriously classified by definitions like $k$-extraction. Rather than encoding the entire sequence, the model learns templates and then reconstructs the sample based on these more broadly applicable patterns. A sequence can thus be perfectly reproduced even if it never appeared during training.

We consider a few templates—stereotyped sequence patterns with a single logical continuation—to define reconstruction candidates. These templates are not intended to be comprehensive, as any stereotyped pattern may permit reconstruction. Our reconstruction candidates are sequences classified as incrementing or repeating by the heuristics described in Section 3.2. As seen in Appendix E, code is more likely to be reconstructed than natural language text. When natural language text is reconstructed, as seen in Appendix F, it often takes the form of a chapter index and it is more likely than code to contain cases of phrase repetition rather than arithmetic sequences.

## 4.3 RECOLLECTION

After excluding highly duplicated recitations and template-based reconstructions, what remains memorized? Despite only seeing a sample a small number of time, the model might still be able to recollect a given sample, although the factors that lead to instant memorization are poorly understood. We consider a sample to be a recollection candidate if it is a candidate for neither recitation nor reconstruction.

Recollected code, seen in Table 4, is largely made up of templating patterns that are not strictly the combination of incrementing and repetition that we use to define templates. The examples of natural language recollection in Table 3 might likewise at first appear to be misclassified recitation cases. Natural language recollection frequently comprises legal or liturgical texts, which would be expected to appear frequently throughout the corpus.

One might conjecture that these sequences are cases of retokenization, i.e., the particular token sequence is rare but the same string is heavily duplicated in the corpus under different tokenizations. However, the dependency tests in Appendix B contradict this hypothesis: the correlation between textual match count and memorization is consistently neutral or negative for recollection candidates. In other words, a rare token sequence is *less* likely to be memorized, not more, if it is a different tokenization of a common string. We instead conjecture that the model appears to memorize slight differences in translation (liturgical text) or indexing (legal) between each variation on a sequence.

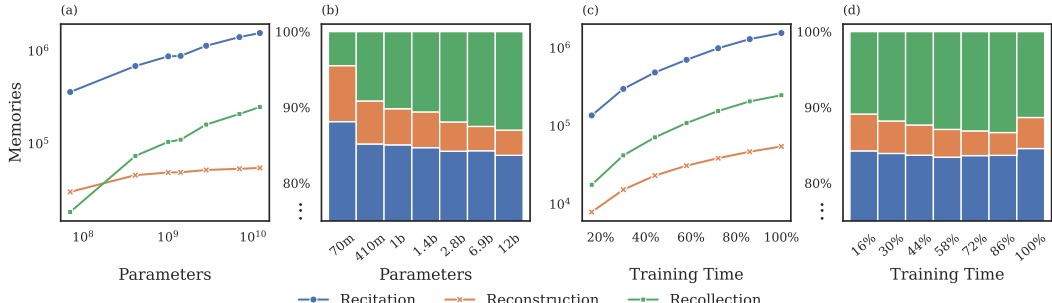

Figure 4: The quantity of memorized data categorized by taxonomy across parameter size and training time. For fully trained models of varying parameter sizes, we give (a) total counts and (b) proportion of memorized samples by category. For the 12B parameter model, we consider intermediate checkpoints during training, also providing for each checkpoint the (c) total memorized counts and (d) proportion of memorized samples by category. ***Note that the proportional plots are truncated at 80%, as recitation is consistently a majority of the overall memorized data.***

## 5 DISTRIBUTION ACROSS SCALE AND TIME

Larger models memorize more data (Biderman et al., 2023a; Carlini et al., 2023; Tirumala et al., 2022), likely because they have more parameters with which to recreate those sequences. Recent

work on deduplication (Sorscher et al., 2022) has argued that larger models are more distorted by duplication, potentially because heavily duplicated sequences are more likely to be memorized (Lee et al., 2021).

Likewise, models memorize more data as training progresses (Tirumala et al., 2022), but it is not known whether the accumulation of memorized examples is caused solely by increased exposure to heavily duplicated samples or whether other factors eventually cause memorization of rare sequences. In this section, we study the impact of training time and model size on each category of memorization.

## 5.1 Model Size

Fig. 4(a) reports the number of examples memorized by each fully trained model, confirming that memorization increases with parameter count. While all types of memorization increase with model size, some increase faster than others. Recollection grows the most (Fig. 4(b)) from 4.49% of the examples memorized in the 70M model, to 11.34% in the 12B model. This disproportionate growth suggests larger models tend to memorize rarer sequences that cannot be trivially reconstructed. Meanwhile, reconstruction barely increases, indicating the smallest models have learned to extrapolate repeating and incrementing templates almost as effectively as the largest.

## 5.2 Time

Over the course of training, LMs are known to memorize an increasing pool of the training data (Tirumala et al., 2022). However, is the cumulative effect due solely to exposure to more memorizable sequences? Due to repeated exposure to the same heavily duplicated data? Or is some structural property of the later model more amenable to exact memorization? To understand why memorization accumulates throughout training, we measure each taxonomic category in intermediate checkpoints for the 12B parameter Pythia model. We find that accumulated memorization cannot be ascribed solely to the number of available samples to memorize or to repeated exposure to highly duplicated samples.

First, in Figure 4(c), we see that models do not simply accumulate memorized samples with a uniform probability through training since memorization increases sub-linearly. Second, if memorization accumulates solely due to repeated exposure to each duplicated sample, recitation of these highly duplicated samples would be the main source of increasing memorization. Instead, the proportion of recitation decreases relative to the amount of memorization (Fig. 4(d)). Therefore, the additional memorization cannot be due to repeated exposure to recitation candidates. Instead, again the largest proportional increase among all categories is in the recollection category. This trend holds until approximately 86% of total training time, which sees a sudden increase in reconstruction. We conjecture that this increase represents a breakthrough in generalizing more complex templates but leave further investigation to future work.

Having considered and rejected both exclusive explanations, we must presume that memorization continues to occur late in training through a combination of repeated exposure, opportunities for memorizing new sequences, and other unexplored factors that may be the focus of future work.

## 6 Predicting memorization

What makes a taxonomy useful, or a reflection of natural kinds? Our position is that categories should differ in the dependencies between features of interest. The most obvious example of validated natural kinds is the case of Simpson's Paradox (Simpson, 1951), a statistical phenomenon in which a pair of variables are correlated across a population, but the direction of correlation reverses when considering each subpopulation category separately. Simpson's Paradox is only the most obvious evidence for natural kinds, but large changes in correlation may support categorical differences even if that correlation does not change direction.

We measure a number of categorical differences in dependencies, including sign and significance differences, in Appendix B. If our intuitive taxonomy did not reflect meaningful differences with respect to the factors in Section 6.3, their dependencies would not differ significantly. We instead find significant differences through statistical tests, suggesting the taxonomy expresses some natural kinds.

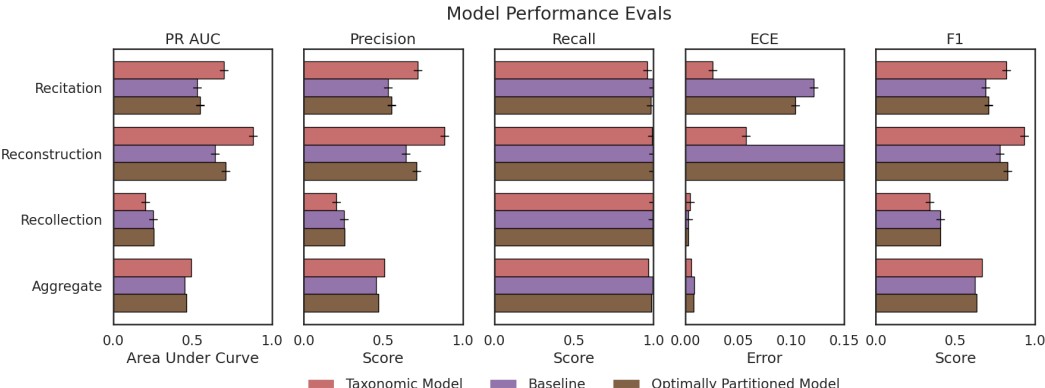

Figure 5: Performance of baseline, proposed taxonomy and optimally partitioned models against various metrics on subsets of test dataset. Confidence interval is standard deviation computed by bootstrapping.

Not only do these differences support our taxonomy as an ontology, but they suggest our taxonomy can help predict memorization from dependent factors. We therefore test the applicability of our taxonomy by creating a predictive model based on the intuitive taxonomic model. Our predictive model divides sequences according to which memorization category they are candidate sequences for, and then uses that category information when predicting the likelihood that the given sequence is memorized. We compare it with a generic baseline model lacking a taxonomy and with a model that uses an automatically selected optimal partition, finding that our taxonomic model supports more accurate predictions.

## 6.1 MODELS

Each model is a logistic regression trained with L2 regularization, a bias parameter, and balanced class weights. We split the representative sample into test and train sets. We then combine the train set with the full memorized sample, reserving a portion as a validation set. For each set, continuous features are normalized to zero mean and unit variance.

**Generic baseline model** The generic baseline is a logistic regression model trained to predict whether a sample is memorized given the features from section 3. It is trained on the training split of the entire memorized dataset and the entire representative Pile sample.

**Intuitive taxonomic model (Ours)** The predictive model based on our intuitive taxonomy is made up of a set of three binary logistic regression models. We divide samples into taxonomic groups before training a separate regression on each taxonomic category.

**Optimally partitioned model** To demonstrate that our intuitive taxonomic model is not simply benefiting from having more degrees of freedom, we devise an equally complex—that is, with the same architecture of three binary logistic regressions—alternative taxonomic model. To provide a strong baseline, we search for a partition based on a set of possible feature-threshold combinations. We train predictive models with the same three-regression architecture as our intuitive taxonomic model, but partitioning based on each feature-threshold combination. The optimal partition is that which supports the best predictive model, which we find categorizes samples based on Huffman coding length followed by sequence duplicate count.

For a given feature, we consider the 25th, 50th and 75th percentiles of the value distribution distribution as potential thresholds. Each feature-threshold pair provides a possible partition split; we select the optimal three-category partition based on F1 score on the aggregate representative test set. Note that our "optimal" partition may not explore our intuitive taxonomy as an option because the threshold search is limited to each feature's quartile values. Our intuitive taxonomy may—and does—therefore outperform the optimal partition.

## 6.2 How good is our taxonomy?

To test our intuitions, we compare our proposed taxonomy to the homogeneous baseline and to our optimal partition. As seen in Fig. 5, the greedy-optimal partition outperforms the aggregate baseline slightly on most metrics, but our intuitive taxonomy is better calibrated and more accurate except on the recollection set, where it has low precision. We conclude that our intuition has guided us to a better taxonomy than searching possible data partitions.

## 6.3 Categorical differences

Having confirmed the benefits of separately considering these three taxonomic categories, Fig. 6 shows how they differ through the feature weights from our regression models.

Recollection candidates—that is, rare sequences—are more likely to be memorized if they have no rare tokens. We posit that there is more resistance to memorizing rare tokens within a sequence, as their prior probability is low.

Meanwhile, the more duplicates a recollection candidate has, the more likely it is to be memorized, whereas recitation candidates are hardly affected by duplicate count. These results suggest that beyond the 5-duplicate threshold, greater exposure hardly leads to memorization.

Another notable difference is in the effect of perplexity: while predictable continuations are strongly associated with memorization across all categories, unpredictable prompts are strongly associated with memorization except for cases of reconstruction. The clear explanation is that high-perplexity prompts often *only* occur as prelude to the same continuation, providing a unique index for the memorized sequence, but that a low-perplexity prompt may also initiate a common template, enabling reconstruction.

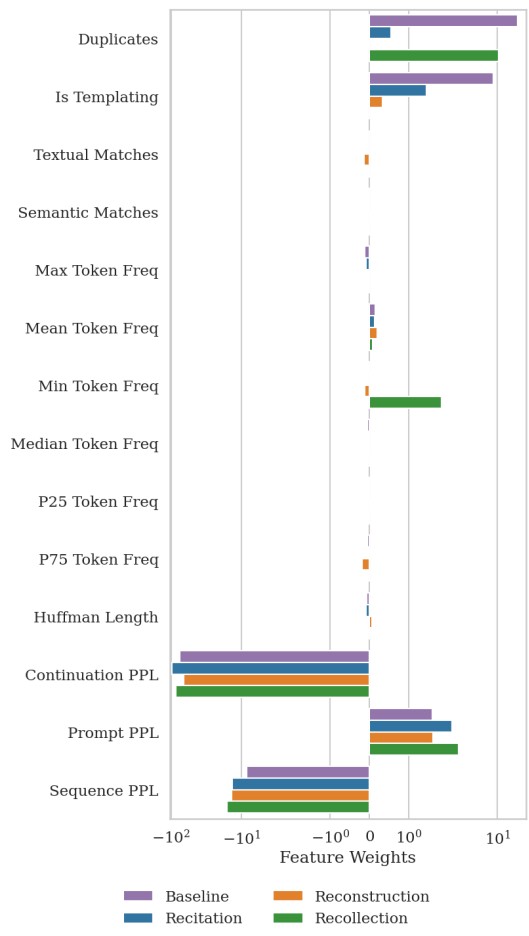

Figure 6: Feature weights from predictive models trained on the homogeneous aggregate baseline and the intuitive taxonomy categories.

## 7 Discussion and future work

We have established that an intuitive taxonomy can be used to improve understanding of memorization. We now relate our methods to the existing literature on memorization and to possible future directions.

## 7.1 Ontologies of memorization

Our work is strongly related to several recent efforts to develop an ontology of memorization. Dankers et al. (2023), studying machine translation, focus primarily on the influence of a sequence during training rather than on the semantics or properties of an individual sequence. They investigate the factors that influence counterfactual memorization, a type of memorization likely to dominate "recollection" cases. They find that rare tokens, long sequence lengths, and high BPE segmentation rate are correlated with counterfactual memorization (Zhang et al., 2021); of these, we only consider rare tokens, which we confirm to predict recollection in particular.

Hartmann et al. (2023) consider what facets of memorization are likely to be relevant to different targets, just as we discuss the differences between motivations grounded in copyright infringement and privacy. Bansal et al. (2023) consider two different kinds of memorization: heuristic memorization, i.e., shortcut learning, and example memorization. Our work focuses on what they call example memorization, further decomposing that category. We do not test their result that high-entropy features can indicate example memorization, but like us, they use this factor to differentiate between their memorization categories.

## 7.2 MEMORIZATION AND TRAINING TIME

Our work fits into an existing literature on how time and scale affect memorization. Biderman et al. (2023b) find that the position of a sequence in training does not affect its likelihood of being memorized, and that smaller models fail to memorize even when repeatedly exposed to a term. Tirumala et al. (2022) find that larger models memorize more training data and forget less during training. They also observe that models memorize nouns and numbers first, using these entities as unique identifiers for individual samples. Our work further expands our understanding of scale in memorization by highlighting that rare sequences compose the fastest-growing category of memorization.

## 7.3 WHICH CATEGORIES DO WE CARE ABOUT?

The relevance of each category depends on our motivation for studying memorization.

1. **Intellectual property violations:** The content most relevant to concerns about intellectual property may be highly duplicated data, such as frequently excerpted passages from a popular book. However, some rare sequences may also be memorized, making recollection potentially relevant to issues of copyright infringement.
2. **Privacy:** If the primary motivation is preventing the memorization of personally identifying information, we may focus on recollection, as issues may arise if a model generates such information even after even a small number of exposures.
3. **Scientific understanding of generalization:** Work like Henighan et al. (2023b) and Bartlett et al. (2020) points to eventual generalization as a result of memorization dynamics. A deeper understanding of these phenomena might focus on reconstruction, which exposes a direct link between apparent overfitting and general pattern recognition.

## 7.4 ONTOLOGIES AND STATISTICS

This taxonomy may serve as an example for future methods of interpreting complex phenomena, in deep learning and elsewhere. We have, in particular, quantified the validity and usefulness of such a taxonomy by comparing predictive models which treat memorization in aggregate to models which treat memorization as a multifaceted phenomenon with our taxonomy. We provide evidence for the taxonomic model by measuring the improvement in predictive judgments when reflecting the dependent and nonlinear thresholded relationship between memorization and the properties that define each taxonomic category.

In future work, we hope that interpretable and useful ontologies can be validated by a similar approach. Our proposal for what makes a good taxonomic model is not only applicable to memorization or even to deep learning phenomena. Instead, by studying interactions and nonlinearities in arbitrary settings, researchers may find complex dependencies and artifacts like Simpson's paradox.

## LIMITATIONS

Our primary goal is to intuitively describe the memorization behavior with a taxonomy and consequently use that taxonomy to investigate how several dominant factors in memorization interact with each other. A secondary goal is to provide an example of how an ontology can be constructed and tested *in general*, as tested with our predictive models. However, these predictive models are not measurements of statistical dependency in general, instead only focusing on linear dependence. Although more general statistical dependencies are studied in the supplementary experiments of

Appendix B, the experiments in the main body of the paper assume linear dependence and so the interacting factors should be evaluated in the context of our supplementary dependency experiments. We believe ontological work inspired by our approach could improve on our work by incorporating more general dependencies.

Another limitation is our definition of memorization. The choice of 32-elicitation has a number of disadvantages, one of them being that we lose a notion of fuzzy or partial memorization, which is considered important in some contexts. Arguably, under a counterfactual memorization definition, we may not see substantial patterns of either recitation or reconstruction. The measurement of memorization is a large area of research with many possible definitions to choose from (Carlini et al., 2022a; Tirumala et al., 2022; Kandpal et al., 2022; Zhang et al., 2021; Zhao et al., 2022; Stock et al., 2022; Schwarzschild et al., 2024).[3]

### ACKNOWLEDGMENTS

This work was enabled in part by a gift from the Chan Zuckerberg Initiative Foundation to establish the Kempner Institute for the Study of Natural and Artificial Intelligence. We would like to thank EleutherAI and CoreWeave for providing the computing resources used in this paper.

We thank Demba Ba for discussion that informed this work.

### AUTHOR CONTRIBUTIONS

*USVSN Sai Prashanth* built sampling infrastructure for the memorized and representative datasets; designed and trained the classifier that determined whether a given sequence was code or natural language; engineered the predictive features for duplicate count, token frequency, and templating; helped optimize perplexity measurement for efficiency; debugged, expanded, and refactored the predictive model training pipeline; generated and modified a number of plots; and helped write.

*Alvin Deng* built the data processing pipeline, designed the predictive model training and evaluation pipelines, analyzed classifier performance on code vs. natural language, generated exploratory visualizations, and helped write.

*Kyle O'Brien* conducted the initial literature review, implemented the pipeline for calculating perplexity, designed several figures including the explanatory diagram, and helped coordinate and manage the project.

*Jyothir S V* conceived the initial idea while conducting early experiments to investigate various memorization patterns and their characteristics. He also engineered the compressibility, semantic match, and textual match features.

*Mohammad Aflah Khan* helped engineer the templating features, visualized and analyzed data, and helped write.

*Jaydeep Borkar* conceived, coded, and visualized potential predictive features for memorization

*Christopher A. Choquette-Choo* ran early experiments that shaped the project.

*Jacob Ray Fuehne* assisted in exploring features by labelling data and feature engineering.

*Stella Biderman* offered high-level guidance, material resources, and help with writing.

*Tracy Ke* supervised and advised statistical testing, especially the dependency tests.

*Katherine Lee* supervised and advised the project and helped with paper writing.

*Naomi Saphra* supervised and advised the project and led paper writing.

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

# A    IMPLEMENTATION OF METRICS

We now provide details of the implementation for metrics considered as potential factors for memorization.

## A.1    NUMBER OF EXACT DUPLICATE SAMPLES

We compute the number of exact duplicates as a 3-step process:

1. For every 32-gram window in the data point $S$ (comprised of 2049 tokens) we compute a rolling hash and store it, along with the window's index and offset in a parallelized set of data frames.

2. For every index position, we compute the same hash for all S[32: 64] (sequence continuations) and store all (index, offset, hash) tuples if their hash is one of the computed sequences continuation hashes.

3. Now, for every sequence continuation, we look at all 32-gram windows with the same hash and compute their number of duplicates as the number of equivalent (same set of tokens in the same order) samples.

The hash function used is similar to Rabin-Karp's rolling hash algorithm. Specifically, consider a token sequence of 32 tokens.

$$S = [c_1, c_2, c_3, ...c_{32}]$$

Let us define two primes $P = 60013$ and $MOD = 10^{18} + 3$. We define their hash function to be

$$H(S) = (c_1 + c_2 * P + c_3 * P^2 + ... + c_{32} * P^{31})\% MOD$$

## A.2    TOKEN FREQUENCY

Token frequencies are calculated across the Pile. For every sequence continuation, we consider the maximum, minimum, median and quartile frequencies of tokens.

## A.3    COMPRESSIBILITY

We use Huffman Coding length to measure how easily a sequence can be compressed. Compressibility provides a rough generalization of internal repetition, where only a few exceptions to some simple repetition pattern might need to be memorized. However, unlike straightforward repetition templates, compressible sequences may not be considered to be reconstructed by the model. Instead, we include compressibility as a filter to evaluate whether LLMs memorize samples that are easier to compress into their parameters.

### A.4 INCREMENTING AND REPEATING TEMPLATES

### A.4.1 INCREMENTING TEMPLATES

To check for an incrementing sequence, we perform the following steps:

- Split the text by whitespace and convert any splits which are numerals in non-decimal bases (e.g., hexadecimal) into base 10.
- Remove escape sequences.
- Within each string, separate contiguous numeric characters from anything else. If two contiguous numeric characters are separated by a period, combine them into their floating point representations.
- Discard if there are fewer than 3 potential numerals in the sequence.
- Check if the sequences are incrementing or repeating.

### A.4.2 REPEATING TEMPLATES

We perform the following steps to check for repeating sequences:

- Obtain a sequence by splitting the text by character.
- Check if the sequences are incrementing or repeating.

We perform the following steps to determine if a sequence generated from either of the above steps is incrementing or repeating.

- For every *templating length*, defined to be less than half length of splits, and for every *position* less than *templating length*, we iterate through splits with start position as *position* and step size set to *templating length*. We then determine if the current iteration is repeating or incrementing.
- For example, if position is 1 and *templating length* is 5, we iterate through positions $[1, 6, 11, 16...]$. if our input splits length is 10, we iterate for all *templating lengths* 1 through 5 and for all positions less than current *templating length*
- Within each iteration, we check:
  - If the current iteration has both texts and numerals, it is neither incrementing nor repeating.
  - If the current iteration has only texts, we consider the current iteration to be repeating if all elements in the iteration are the same.
  - If the current iteration has only numerals, we consider current iteration to be incrementing if all the numerals are in an arithmetic progression. If the difference in AP is 0, we consider it to be repeating instead.
- Input splits are considered as repeating if all iterations for a given *templating length* are repeating.
- Input splits are considered as incrementing if atleast one of the iterations for a given *templating length* are incrementing and others, for the same *templating length*, are either incrementing (or) repeating.
- For all templating lengths, if any length of them has been found to be incrementing or repeating, we return *True* (corresponding to the fact that the text is indeed a template) and *diff*.
- Note that, in the case of sequences generated while checking for a repeating template, we do not have any numerals.

## B DEPENDENCY TESTS FOR INFLUENCE OF FEATURES ON MEMORIZATION

This section contains visualizations of various dependency tests between memorization likelihood and our target features. We look at dependencies on code (Fig. 9), natural language (Fig. 8), and both

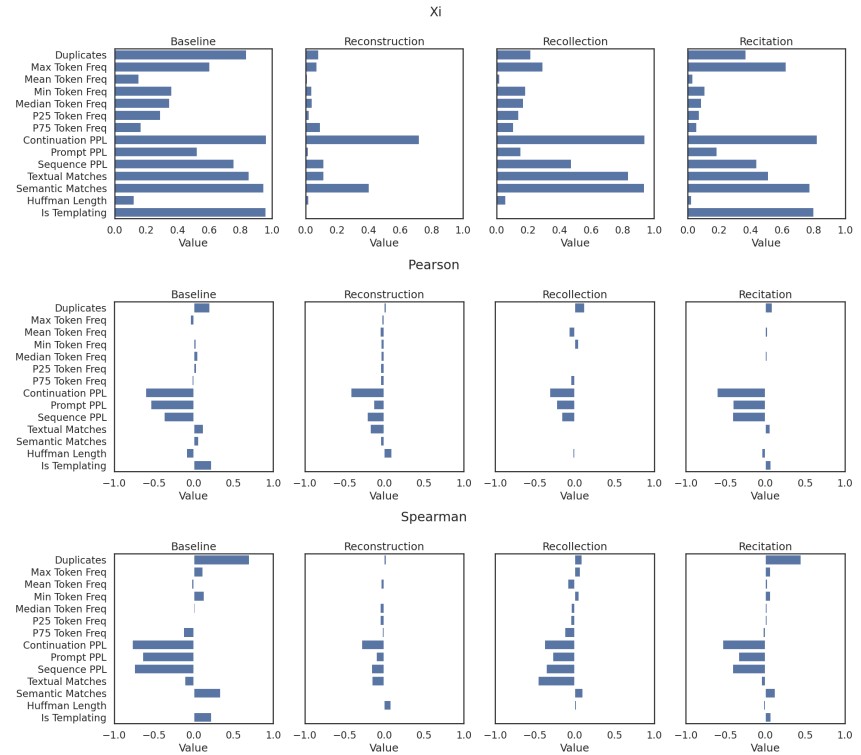

Figure 7: Dependency measurements between influence factors and memorization.

(Fig. 7). These tests are more general and have stronger guarantees than simply looking at regression weights, which have a number of flaws. For example, we see in Fig. 6 that regression can reallocate bias terms to features that take on a consistent value, giving them spurious weight.

## C TABLES FOR SCALING EXPERIMENTS

Figure 4 visualizes the count and proportion of memorized samples by category across time and scale. We present the raw statistics for each taxonomic category across model size in Table 1 and training time in Table 2.

| | Recitation | | Reconstruction | | Recollection | |
| --- | --- | --- | --- | --- | --- | --- |
| Model | Count | Percent | Count | Percent | Count | Percent |
| 70m | 362,550.0 | 88.12% | 30,430.0 | 7.40% | 18,468.0 | 4.49% |
| 410m | 690,726.0 | 85.17% | 46,076.0 | 5.68% | 74,238.0 | 9.15% |
| 1b | 878,456.0 | 85.05% | 49,253.0 | 4.77% | 105,163.0 | 10.18% |
| 1.4b | 887,549.0 | 84.68% | 49,435.0 | 4.72% | 111,120.0 | 10.60% |
| 2.8b | 1,141,180.0 | 84.21% | 52,416.0 | 3.87% | 161,620.0 | 11.93% |
| 6.9b | 1,416,014.0 | 84.27% | 53,968.0 | 3.21% | 210,314.0 | 12.52% |
| 12b | 1,566,369.0 | 84.56% | 55,114.0 | 4.10% | 249,733.0 | 11.34% |

Table 1: The number of memorized samples for each taxonomic category across model size. These results are visualized in Figure 4(a) and 4(b).

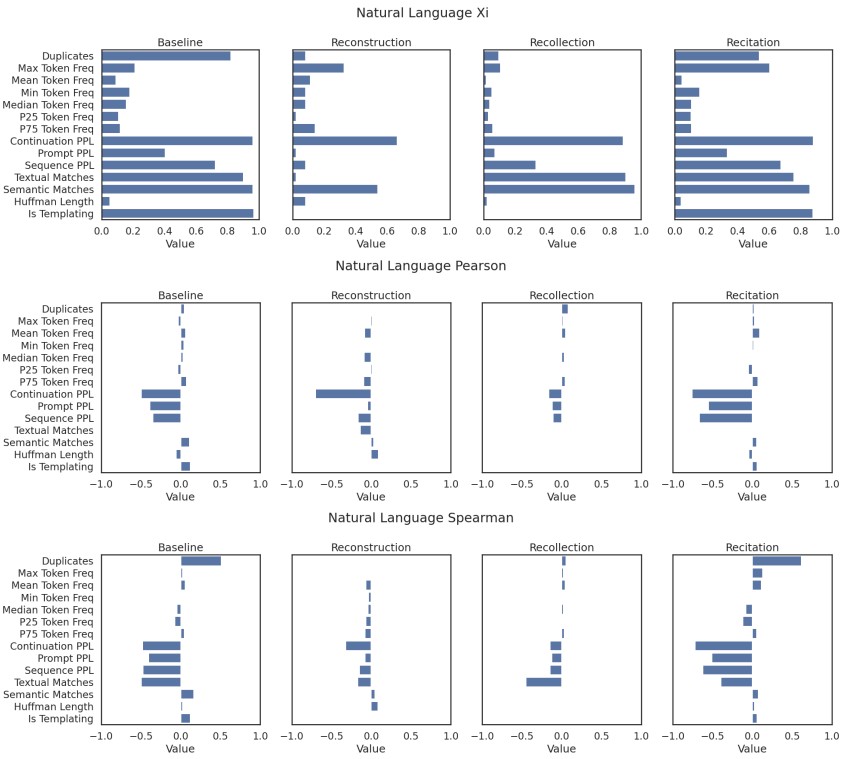

Figure 8: Dependency measurements between influence factors and memorization for natural language samples.

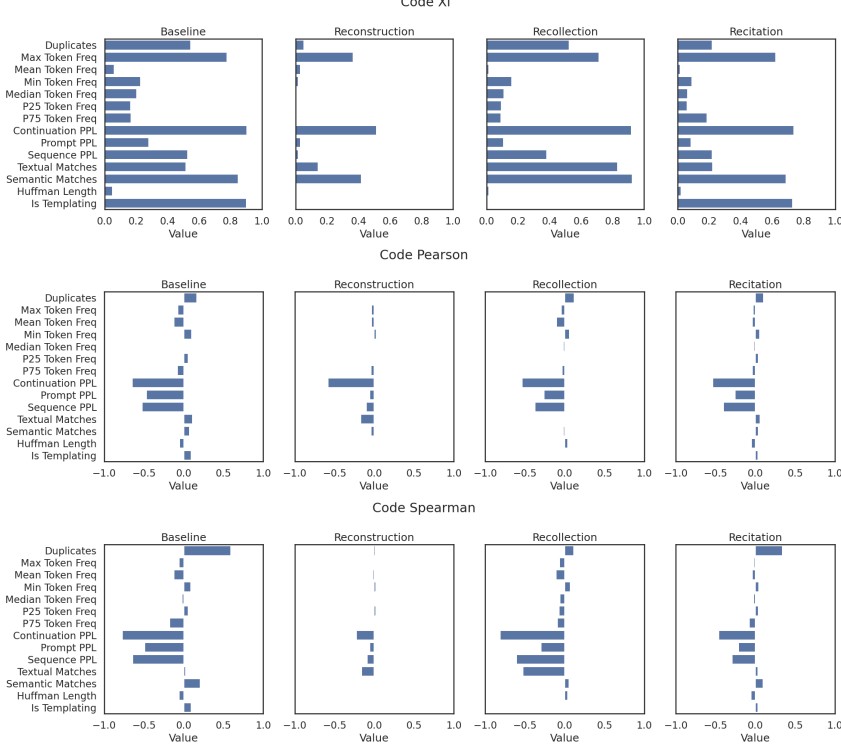

Figure 9: Dependency measurements between influence factors and memorization for code samples.

| Checkpoint | Recitation | | Reconstruction | | Recollection | |
|---|---|---|---|---|---|---|
| | Count | Percent | Count | Percent | Count | Percent |
| 16% | 137,681.0 | 84.25% | 7,960.0 | 4.87% | 17,777.0 | 10.88% |
| 30% | 301,146.0 | 83.92% | 15,379.0 | 4.29% | 42,338.0 | 11.80% |
| 44% | 489,629.0 | 83.69% | 23,333.0 | 3.99% | 72,105.0 | 12.32% |
| 58% | 710,823.0 | 83.42% | 31,260.0 | 3.67% | 109,985.0 | 12.91% |
| 72% | 999,867.0 | 83.63% | 38,999.0 | 3.26% | 156,712.0 | 13.11% |
| 86% | 1,308,538.0 | 83.66% | 47,145.0 | 3.01% | 208,372.0 | 13.32% |
| 100% | 1,566,369.0 | 84.56% | 55,114.0 | 4.10% | 249,733.0 | 11.34% |

Table 2: The number of memorized samples for each taxonomic category across training time for Pythia 12b. 14,000 is the final checkpoint. These results are visualized in Figure 4(c) and 4(d).

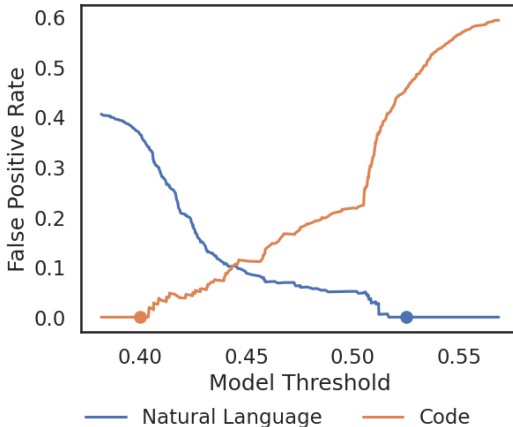

Figure 10: False positive rates across various thresholds on randomly sampled sequences of Pile. We choose $\leq 0.4$ as threshold for determining code samples and a threshold of $\geq 0.525$ for determining natural language

## D  CLASSIFYING EXAMPLES AS NATURAL LANGUAGE OR CODE

To train a Natural Language vs Code classifier, we fine-tune DistilBert (Sanh et al., 2020) on uniformly random sampled Bookcorpus (Zhu et al., 2015) and github-code datasets. We train it with learning rate of $10^{-7}$ and batch size of 256 for a total of 1000 steps and observe validation f1 score of $0.9950$ on a held of evaluation set.

To select an optimal threshold for this classifier on memories dataset, we randomly sample 500 sequences and manually label them. To make sure that precision is high for our models, we choose $\leq 0.4$ as threshold for determining code samples and a threshold of $\geq 0.525$ for determining natural language samples, based on the points marked in Figure 10, which mark points of near 100% precision for classifying each category.

## E  LIKELIHOOD OF MEMORIZATION FOR CODE AND NATURAL LANGUAGE

We study the likelihood that a sample that has been confidently classifier as code or NL (Appendix D) is memorized across time and scale. For example, for all samples confidently classified as code Figure, what is the proportion of samples which are memorized?

Figure 11 shows that code samples are more likely to be memorized than NL across categories. This trend suggests that certain intrinsic factors about code make it more susceptible to memorization, even for recollection samples where memorization cannot be attributed to obvious patterns and

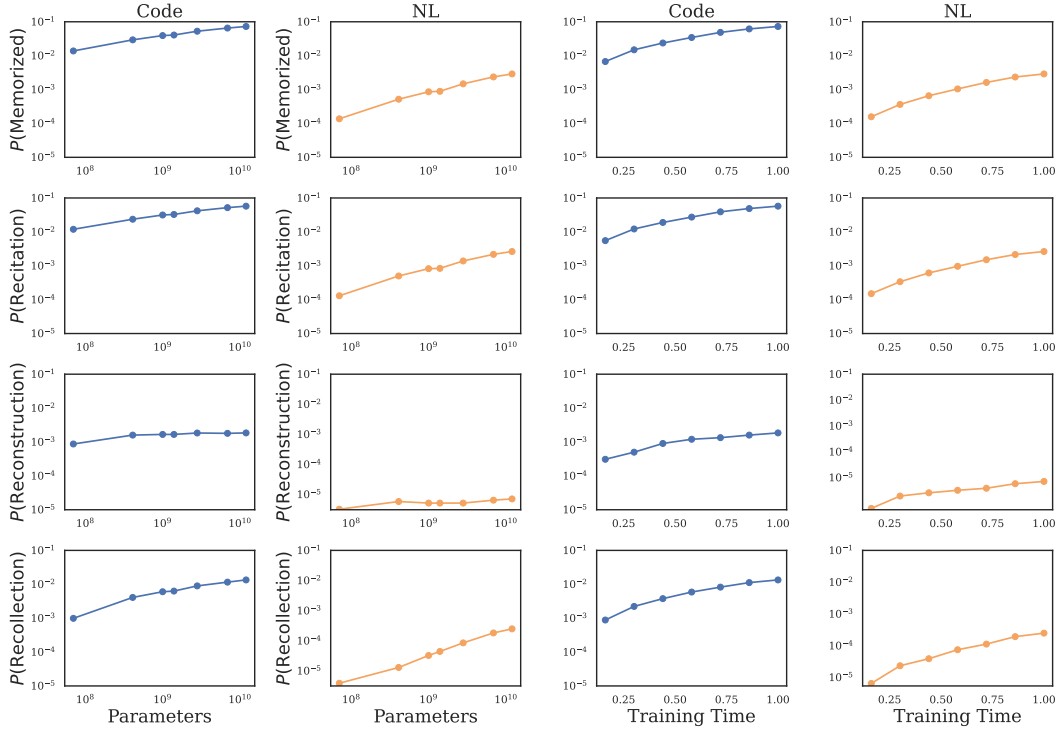

Figure 11: We study how likely models are to memorize samples confidently classified as code or NL. We calculate the likelihood for each distribution (code vs. NL) separately. Figures include probability across a model scale and training time. Models memorize a greater proportion of code samples than NL across all categories, model scale, and training time.

high duplication. Both code and NL become more likely to be memorized across scale, except for reconstruction samples, which remain comparatively unchanged.

## F  EXAMPLES OF MEMORIZED CONTINUATION SEQUENCES

Table 3 provides examples of memorized natural language text in each memorization category and Table 4 provides examples of memorized code. Samples are classified using the methodology in Appendix D.

## G  ALTERNATIVE RECITATION THRESHOLDS

We have selected $> 5$ as our duplication threshold for categorizing a sequence as a recitation candidate, based on the analysis in Fig. 3. While we have shown that our resulting taxonomy outperforms those based on possible quartile cutoffs, we have not compared it to other small thresholds. In Fig. 12, we perform this comparison and find that the threshold we selected based on intuitions is at least as good as a similar but smaller (>1) or larger (>10) threshold.

| Category | Text | Count |
|---|---|---|
| Recitation | -11 NASB
Or do you not know that the unrighteous will not inherit the kingdom of God? Do not be deceived; neither fornicators, nor idolaters, nor adulterers, nor effeminate, nor homosexuals, nor thieves, nor the covetous, nor drunk | 175 |
| Recitation | 2d 1234, 1238 (8th Cir.1990). On the other hand, the Federal Rules of Civil Procedure have authorized for nearly 60 years "motions for summary judgment upon proper showings of the lack of a genuine, triable issue of material fact." Celotex Corp. v. Catrett | 86 |
| Recitation | view of life, food, cocktails, fitness, and fun.
This blog is just a regular guy's view of life, food, cocktails, fitness, and fun. My opinions, musings, observations, rantings, ravings, foodie adventures, and overall humorous pontification of | 52 |
| Recitation | 000,
from the tribe of Simeon 12,000,
from the tribe of Levi 12,000,
from the tribe of Issachar 12,000,
from the tribe of Zebulun 12,000,
from the tribe of Joseph 12,000, | 7 |
| Reconstruction | THEIR EAGLE ONAND THE DIRTY BIRDS READY AS WELL DOWN AND GET THEIR EAGLE ONAND THE DIRTY BIRDS READY AS WELL DOWN AND GET THEIR EAGLE ONAND THE DIRTY B | 4 |
| Reconstruction | 181 \|182 \|183 \|184 \|185 \|186 \|187 \|188 \|189 \|190 \|191 \|192 \|193 \|194 \|195 \|196 \|197 \|198 \|199 \|200 \|201 \|202 | 2 |
| Reconstruction | 1970–71 Turkish Third Football League season
Promotion and relegation:
1971–72 Turkish Third Football League season
Promotion and relegation:
1972–73 Turkish Third Football League season
Promotion and relegation:
1973–74 Turkish Third Football | 1 |
| Reconstruction | , 28-63, 28-64, 28-65, 28-66, 28-67, 28-68, 28-69, 28-70, 28-71, 28-72, 28-73, 28-74, 28-75, 28-76, 28-77, 28-78 | 3 |
| Recollection | affect the child;
¶70 (b) The wishes of the child, as expressed directly by the child or through the child's guardian ad litem, with due regard for the maturity of the child;
¶71 (c) The custodial history of the child, | 2 |
| Recollection | will be too late!
" 24 "Strive to enter through the narrow door. For many, I tell you, will seek to enter and will not be able. 25 When once the master of the house has risen and shut the door, and you begin to stand outside and to knock at the door, | 4 |
| Recollection | , and freedom.Revava
Revava (), is an Orthodox Jewish Israeli settlement in the West Bank. Located between Barkan and Karnei Shomron, it falls under the jurisdiction of Shomron Regional Council. In it had a population of.
The international community considers Israeli settlements in | 2 |
| Recollection | § 2254(d)).
A state-court decision is considered "contrary to... clearly established Federal law" if the two are "diametrically different, opposite in character or nature, or mutually opposed." Williams v. Taylor, 529 U.S. 362, 405 ( | 5 |

Table 3: Examples of natural language (classified per Appendix D) from each memorization category.

| Category | Text | Count |
|---|---|---|
| Recitation | >-task"> <a class="nav-group-task-link" href="../Extensions/Int.html">Int</a> </li> <li class="nav-group-task"> <a class="nav-group-task-link" href="../Extensions/ | 5310 |
| Recitation | > <widget class="GtkButton" id="entCleanBut"> <property name="label" translatable="yes">... :</p> <ul> <li> <p>Must be one of: true, false, 1, 0.</p> </li> </ul> | 3689 |
| Recitation | = null, CancellationToken cancellationToken = default(CancellationToken)) { if (Client.SubscriptionId == null) { throw new ValidationException(ValidationRules.CannotBeNull, "this.Client.SubscriptionId"); } if | 227 |
| Recitation | .Object</h3> equals, getClass, hashCode, notify, notifyAll, wait, wait, wait</li> </ul> </li> </ul> </li> </ul> </div> <div class="details"> | 18963 |
| Reconstruction | FileEntry("/base1/dir1/",fe, age); fe.name = "file3"; fi -> updateFileEntry("/base1/dir1/",fe, age); fe.name = "file4"; fi -> updateFileEntry("/base1/ | 2 |
| Reconstruction | ="time2[]" value="5" ></td> <td><input type="checkbox" name="time2[]" value="6" ></td> <td><input type="checkbox" name="time2[]" value="7" ></td> < | 2 |
| Reconstruction | XMM3 (1ULL « 28) #define DBG_CTX_EX_PART_FLAG_XMM4 (1ULL « 29) #define DBG_CTX_EX_PART_FLAG_XMM5 (1ULL « 30) #define DBG_ | 2 |
| Reconstruction | " /> <Compile Include="Message\MFN_M06.cs" /> <Compile Include="Message\MFN_M07.cs" /> <Compile Include="Message\_M08.cs" /> <Compile Include="Message\MFN_ | 3 |
| Recollection | OFFSET + (2 * FPREG_SIZE)) #define PROBE_CPU_Q3_OFFSET (PROBE_FIRST_FPREG_OFFSET + (3 * FPREG_SIZE)) #define PROBE_CPU_Q4_OFFSET (PROBE_FIRST_FPREG_OFFSET + | 2 |
| Recollection | ); } uint16_t WS2812FX::mode_custom_5() { return customModes[5](); } uint16_t WS2812FX::mode_custom_6() { return customModes[6](); } uint16 | 2 |
| Recollection | XMMM128, __) /* 0xDC */ NORMAL("paddusb", MM_XMM, MMM64_XMMM128, __) /* 0xDD */ NORMAL("paddusw", MM_XMM, MMM64_X | 2 |
| Recollection | DBF3B /* icon1.png */; }; 651A5A7E177AE2D8003DBF3B /* icon2.png in Resources */ = {isa = PBXBuildFile; fileRef = 651A5A7C177AE2D8003DBF | 2 |

Table 4: Random examples of code (as classified per Appendix D) from each memorization category.

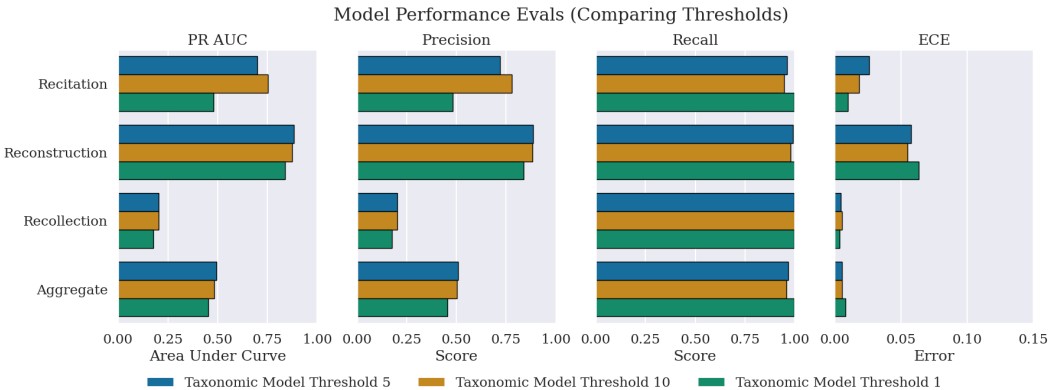

Figure 12: Comparison of memorization predictor performance, similar to those trained in Section 6. Thresholds at 1 or 10 do not generally outperform our selected threshold of 5.

