# OpenReview forum: "Recite, Reconstruct, Recollect: Memorization in LMs as a Multifaceted Phenomenon"
_ICLR.cc/2025/Conference — ICLR 2025 Poster_

### Official Review · Reviewer_p6LS · 2024-10-26

**Soundness:** 3
**Presentation:** 2
**Contribution:** 3
**Rating:** 8
**Confidence:** 4

**Summary:**

This paper proposes a taxonomy of memorization in language models, breaking it down into three categories: recitation of highly duplicated sequences, reconstruction of inherently predictable patterns, and recollection of rare sequences. The authors validate their taxonomy through statistical analysis and predictive modeling, showing that different factors influence memorization differently across categories.

They analyze how memorization patterns evolve with model scale and training time, finding that recollection grows disproportionately with model size. The paper illustrates the practical value of this taxonomy by showing how different categories matter for different applications (privacy, copyright, scientific understanding) and by achieving better predictive performance compared to both a baseline without taxonomy and an optimally partitioned model.

**Strengths:**

- **The paper provides a methodologically sound approach to developing and validating taxonomies in ML research (and beyond).** By grounding qualitative distinctions in statistical analysis (e.g., following the example of Simpson's paradox), it offers a template for studying complex phenomena in ML beyond memorization that could be of interest for the ICLR community.
- **The taxonomy enables discoveries about memorization dynamics.** For instance, the finding that duplicate count beyond 5 barely affects recitation likelihood challenges simple assumptions about exposure and memorization. The categorical distinctions also help align research directions with specific applications (e.g., privacy vs. copyright concerns).
- **Analysis of the semantic properties of the sequence.** This type of statistics provides valuable insights into how models can perfectly reproduce sequences through pattern completion rather than pure memorization for the reconstruction category. This distinction, while simple in hindsight, is important for understanding the relationship between memorization and generalization, and the limit of the k-extractability definition of memorization.

**Weaknesses:**

- **The predictive model's performance is relatively poor** (precision ~0.5, PR AUC ~0.5), despite including continuation perplexity as an input feature. This raises questions about the practical utility of the identified factors in predicting memorization. The heavy reliance on continuation perplexity for every category (see Figure 5.), which is closely related to the definition of k-extractability, makes it difficult to assess the independent predictive power of other factors.
- **No clear progress in understanding the memorisation of rare sequence.** While the paper identifies recollection of rare sequences as an important phenomenon, particularly as models scale, it provides limited insight into the underlying mechanisms. This gap is particularly notable given the paper's emphasis on understanding different types of memorization.
- **The presentation lacks clarity at times.**
	- When introducing the taxonomy, early concrete examples of each category would significantly improve understanding.
	- The paper should also better highlight the distinction between intuitive notions of memorization and the technical definition of k-extractability used in the study. This could help the reader understand why the reconstruction phenomenon (where sequence outside of the training set could be predicted perfectly) fall in the scope of the study of memorization.
	- The study could benefit of including reference to a broader set of references such as the study of mechanistic interpretability and training providing more insights on how and when models become able to predict simple sequences. See for instance "In-context Learning and Induction Heads", by Olsson et al.
- **Methodological limitation in the computation of the corpus statistics.**
	- The corpus statistics are not broken down into prompt/continuation/full sequence. This could enable the isolation of sequences with a frequent prompt but infrequent continuation, or the opposite for instance. The paper doesn't clearly state which one of the three is used for the corpus statistics.
	- If I understood correctly, the semantic similarity measurements are made between sequences of length 64 (or 32) tokens (the memorized/non-memorized samples), and the 2049-token sequences from the Pile. This length mismatch could introduce heavy distortion as even if the small sequence is included in the large sequence, it is not clear that the cosine similarity of their embedding would be similar.

**Questions:**

1. How would the predictive model perform without continuation perplexity as a feature? This would help assess how much signal the other identified factors provide to predict memorization.
2. How novel is the introduction of statistically validated taxonomy? Do similar case studies exist in other fields? Beyond the reference to the Simpson paradox, the paper doesn't include references in this area.
3. Are corpus statistics computed on the prompt, continuation or the full sequence?
4. What are the characteristics of sequences with high duplicate counts (>100) that don't get memorized? Understanding these cases might provide insight into factors that inhibit memorization despite repeated exposure.
5. How sensitive are the semantic similarity measurements to the length mismatch between the 64 (or 32)-token samples and the 2049-token sequences they're compared against?
6. (Less important) Could the sudden increase in reconstruction at 86% of training be related to the "induction bump" phenomenon described in the mechanistic interpretability literature? Again, see "In-context Learning and Induction Heads", by Olsson et al. for the introduction of the concept.


Minor points:

The Figure 6 would benefit from being vertically compressed without deforming the text.

Likely typo lines 140-142: "we generate document embeddings for each full sequence using SBERT and count the number of sequences with cosine similarity ≤ 0.8. These sequences are semantically similar but may not be exact token-level duplicates." -> I guess it should be "cosine similarity ≥ 0.8" instead.

---

> ### Author Response · Authors · 2024-11-19
> **Weaknesses**
>
> Weaknesses:
> > The predictive model's performance is relatively poor (precision ~0.5, PR AUC ~0.5), despite including continuation perplexity as an input feature. This raises questions about the practical utility of the identified factors in predicting memorization. The heavy reliance on continuation perplexity for every category (see Figure 5.), which is closely related to the definition of k-extractability, makes it difficult to assess the independent predictive power of other factors.
>
> The goal of our taxonomy is not to directly predict memorization, but to analyze the factors that make it more or less likely. While our taxonomy might be useful in future work on predicting memorization, our linear model allows us to control for various features, especially those that are otherwise dominate other factors, such as perplexity.
>
> > No clear progress in understanding the memorisation of rare sequence. While the paper identifies recollection of rare sequences as an important phenomenon, particularly as models scale, it provides limited insight into the underlying mechanisms. This gap is particularly notable given the paper's emphasis on understanding different types of memorization.
>
>
> Our main finding in the case of rare sequences is that, excluding simple templating behaviors such as repetition and incrementation, most are likely cases of more complex templating behaviors mixed with recitation. In particular, the rare biblical passages and legal texts appear to be high frequency sequences that are either reindexed or have some unusually translated words that can be memorized. We specifically exclude the possibility (based on Dankers et al. (2023)) of memorized rare sequences simply being rare retokenizations of common sequences, by studying surface level string similarity.
>
> > When introducing the taxonomy, early concrete examples of each category would significantly improve understanding.
> > The paper should also better highlight the distinction between intuitive notions of memorization and the technical definition of k-extractability used in the study. This could help the reader understand why the reconstruction phenomenon (where sequence outside of the training set could be predicted perfectly) fall in the scope of the study of memorization.
>
> Thank you for these recommendations. We are currently adding them to a revised version.
>
> > The study could benefit of including reference to a broader set of references such as the study of mechanistic interpretability and training providing more insights on how and when models become able to predict simple sequences. See for instance "In-context Learning and Induction Heads", by Olsson et al.
>
> It is true that repetition could be implemented by mechanisms similar to induction heads. Are there citations to mechanistic work on numerical incrementing that you also feel we could include?
>
> > The corpus statistics are not broken down into prompt/continuation/full sequence. This could enable the isolation of sequences with a frequent prompt but infrequent continuation, or the opposite for instance. The paper doesn't clearly state which one of the three is used for the corpus statistics.
>
> We use the continuation currently, but would happily implement a more granular analysis for the camera ready. (These experiments would take too long for the rebuttal period.)
>
> > If I understood correctly, the semantic similarity measurements are made between sequences of length 64 (or 32) tokens (the memorized/non-memorized samples), and the 2049-token sequences from the Pile. This length mismatch could introduce heavy distortion as even if the small sequence is included in the large sequence, it is not clear that the cosine similarity of their embedding would be similar.
>
> We agree that there would be semantic differences between similar sequences of different lengths, and we do control for that. We specifically look only at semantic similarity between the same length (64 tokens) as the full sequence. We consider every 64-token sequence available from the pile. We clarify this in the updated paper.

---

> > ### Author Response · Authors · 2024-11-19
> > **Questions**
> >
> > > How would the predictive model perform without continuation perplexity as a feature? This would help assess how much signal the other identified factors provide to predict memorization.
> > How novel is the introduction of statistically validated taxonomy? Do similar case studies exist in other fields? Beyond the reference to the Simpson paradox, the paper doesn't include references in this area.
> >
> > As far as we know, a statistically validated taxonomy is not common practice in the science of deep learning literature, but we believe that it is a generally useful approach for interpretation and that we should import these practices from other empirical sciences, e.g., economics. Simpson’s paradox is the common example of when stratification is a valid approach, but as far as we know, our use of classical methods for computing statistical dependencies for interpretation is novel in the analysis of deep learning models.
> >
> > > Are corpus statistics computed on the prompt, continuation or the full sequence?
> >
> > Continuation.
> >
> > > What are the characteristics of sequences with high duplicate counts (>100) that don't get memorized? Understanding these cases might provide insight into factors that inhibit memorization despite repeated exposure.
> >
> > While we don’t do an in-depth qualitative analysis of unmemorized highly duplicated sequences, our controlled analysis does suggest that sequences with many common words (based on our P75 frequency feature) are less likely to be recited, even if highly duplicated.
> >
> > > How sensitive are the semantic similarity measurements to the length mismatch between the 64 (or 32)-token samples and the 2049-token sequences they're compared against?
> >
> > See weaknesses. We specifically look only at semantic similarity between the same length (64 tokens) as the full sequence.
> >
> > > (Less important) Could the sudden increase in reconstruction at 86% of training be related to the "induction bump" phenomenon described in the mechanistic interpretability literature? Again, see "In-context Learning and Induction Heads", by Olsson et al. for the introduction of the concept.
> >
> > Induction heads could certainly be connected to reconstruction, but they form earlier in training. Olsson et al. saw the formation of an induction head in a 2 layer model after 1B tokens; The Pile is 207B tokens, and training is slightly longer than an epoch. While it is possible that somehow this model develops an induction head more slowly, the large quantity of code in the dataset makes it likely that induction heads would instead form more quickly.
> >
> > > The Figure 6 would benefit from being vertically compressed without deforming the text.
> > > Likely typo lines 140-142: "we generate document embeddings for each full sequence using SBERT and count the number of sequences with cosine similarity ≤ 0.8. These sequences are semantically similar but may not be exact token-level duplicates." -> I guess it should be "cosine similarity ≥ 0.8" instead.
> >
> > Thank you for highlighting these presentation issues, we have corrected them in our updated paper.

---

> > > ### Comment · Reviewer_p6LS · 2024-11-23
> > > **Main concern stays the performance of the predictive model without continuation perplexity**
> > >
> > > Thank you for your answers.
> > >
> > > The main limitation I perceive is still the the performance of the predictive model, especially without continuation perplexity as a feature. As argued in my response to the 'Weakness' comment, measuring the performance of the predictive model without continuation perplexity seems to be crucial to assess the relevance of the taxonomy.
> > >
> > > It seems you did not address my question #1 in your answer.
> > > > How would the predictive model perform without continuation perplexity as a feature?
> > >
> > > 1. If the performance turns out poor, the work is still a valuable exploration of how a set of given hand-crafted, interpretable factors influence memorisation. However, this need to be clear that these factors alone are not enough to capture the significant variability of memorisation.
> > > 2. If the performance is significant, the hand-crafted, interpretable factors are sufficient to predict memorisation. This would make the taxonomy a much more significant contribution.
> > >
> > > **Minor discussion points**
> > >
> > > > Induction heads could certainly be connected to reconstruction, but they form earlier in training. Olsson et al. saw the formation of an induction head in a 2 layer model after 1B tokens; The Pile is 207B tokens, and training is slightly longer than an epoch. While it is possible that somehow this model develops an induction head more slowly, the large quantity of code in the dataset makes it likely that induction heads would instead form more quickly.
> > >
> > > Indeed you are right that the timing doesn't seem to match. Maybe this could be an artefact of the cosine learning rate schedule used to train the pythia models.

---

> > > > ### Author Response · Authors · 2024-11-25
> > > > **Predictive model without continuation perplexity**
> > > >
> > > > As requested we redid the analysis for the predictive model removing continuation perplexity (CP) as a feature. The results are as follows:
> > > >
> > > > |                       | PR AUC | Precision | Recall | ECE  |
> > > > |-----------------------|--------|-----------|--------|------|
> > > > | Recitation (Tax)      | 0.61   | 0.62      | 0.98   | 0.03 |
> > > > | Recitation (Base)     | 0.46   | 0.46      | 1.00   | 0.15|
> > > > | Reconstruction (Tax)  | 0.79   | 0.80      | 0.97   | 0.10 |
> > > > | Reconstruction (Base) | 0.68   | 0.68      | 1.00   | 0.15|
> > > > | Recollection (Tax)    | 0.13   | 0.12      | 1.00   | 0.01 |
> > > > | Recollection (Base)   | 0.16   | 0.16      | 1.00   | 0.005 |
> > > > | Agg. (Tax)            | 0.38   | 0.38      | 0.99   | 0.01 |
> > > > | Agg. (Base)           | 0.36   | 0.36      | 1.00   | 0.01 |
> > > >
> > > > *NB: the above numbers are rounded to the hundreds for ease of readability, with the exception of Recollection (Base) on ECE as this would cause it to round to zero. We include that value rounded to the thousands to emphasize that it is not zero. Plots with full error bars will be included in the paper itself, but there are no instances where the 2 S.E. error bars overlap or where the error bars include zero.*
> > > >
> > > > **At a high level, our primary conclusion is that nothing changes in our analysis when excluding the CP.** For every measurement made, if the taxonomic model outperformed the baseline when including the CP it continues to do so when excluding it and vice versa. Additionally, the exclusion of CP influenced every metric in the same direction: if the taxonomic model improved when excluding CP then so did the baseline model and vice versa. While it is a little counter-intuitive that such a thing could happen, this never happened with the aggregate scoring and currently believe this is either reflective of Simpson's paradox or of the changing of the relative difficulty of the different subcategories. That is, some of the points lost when predicting Recitation PR AUC is due to the bits of optimization that had previously been allocated to that being instead allocated to Recollection PR AUC by the optimization algorithm.
> > > >
> > > > We were surprised at how small the changes to these values were, in general. For example, the aggregate PR AUC fell by 0.126 for the taxonimic model and 0.089 for the baseline model. As has been discussed, CP is expected to be the single biggest contributor to these scores and so having it fall by about 25% when CP is excluded points to the irreducible complexity being quite high, or alternatively there being a very long tail of minor but meaningful variables involved. As an additional point of reference, Precision fell by 0.13 (Tax) and 0.092 (Baseline), Recall increased by 0.02 (Tax) and 0.00 (Baseline), and ECE increased by 0.003 (Tax) and 0.003 (Baseline). Note that Recall's small change is due to the Recall already being extremely high and the CP-excluded recall couldn't go up as it was already at 1.00.
> > > >
> > > > **In general we view this as further evidence in support of our model.** It is absolutely the case that there is either a very large amount of irreducible loss or there are meaningful sources of variation that we are not capturing. However the sources of variation we are able to capture seem to encode significant amounts of human-interpretable structure.

---

> > > > > ### Comment · Reviewer_p6LS · 2024-11-25
> > > > >
> > > > > Thank you for the follow up experiments! I agree that these experiments are further evidence supporting your model. However, I realised that the prompt PPL and sequence PPL are still included in the input. If the experiment are fast enough to run, **could you measure the performance without any perplexity as input?**
> > > > >
> > > > > As for the source of irreducible variation, it would be great to include a short discussion on this topics in section 7 or subsection 6.2.
> > > > >
> > > > > All in all, given the discussion period is reaching an end, and that you addressed my main concerns, I will increase my score from 6 to 8.

---

> ### Comment · Reviewer_p6LS · 2024-11-23
>
> Thank you for your comment! It provides insightful clarifications.
>
> > The goal of our taxonomy is not to directly predict memorization, but to analyze the factors that make it more or less likely. While our taxonomy might be useful in future work on predicting memorization, our linear model allows us to control for various features, especially those that are otherwise dominate other factors, such as perplexity.
>
> I understand that the goal is not direct prediction of memorization. However, would you agree that the performance in predicting memorization is a good proxy to estimate how much the factors you introduce capture the memorization phenomenon? In other words, if the prediction performance is poor, it's a sign that there are unexplained variability not contained in the input factors (note that this variability could be irreducible noise, but this needs to be discussed). Conversely, if the prediction performance is good, it means the factors identified capture the main variable that condition memorization.
>
>
> > Our main finding in the case of rare sequences is that, excluding simple templating behaviors such as repetition and incrementation, most are likely cases of more complex templating behaviors mixed with recitation. In particular, the rare biblical passages and legal texts appear to be high frequency sequences that are either reindexed or have some unusually translated words that can be memorized. We specifically exclude the possibility (based on Dankers et al. (2023)) of memorized rare sequences simply being rare retokenizations of common sequences, by studying surface level string similarity.
>
> Thank you for clarifying this contribution. I recognize the value of ruling out a hypothesis from the litterature. However, I maintain that given that recollection is defined as everything that is not recitation nor reconstruction, this category doesn't provides deep understanding of the sequences it contains. 'more complex templating behaviors mixed with recitation' gives a fuzzy intuition (e.g. when is a complex templating behavior so complex/specific that it becomes a recitation?) for what could be in here. It could still be valuable to guide further hypothesis.
>
> > It is true that repetition could be implemented by mechanisms similar to induction heads. Are there citations to mechanistic work on numerical incrementing that you also feel we could include?
>
> I am not aware of work specifically studying numerical incrementing.
>
> > We agree that there would be semantic differences between similar sequences of different lengths, and we do control for that. We specifically look only at semantic similarity between the same length (64 tokens) as the full sequence. We consider every 64-token sequence available from the pile. We clarify this in the updated paper.
>
> Thank you for this clarification. This adresses my concern on this point.

---

### Official Review · Reviewer_FK6z · 2024-11-03

**Soundness:** 3
**Presentation:** 3
**Contribution:** 3
**Rating:** 8
**Confidence:** 3

**Summary:**

This comprehensive paper presents a novel taxonomic analysis of memorization in LLMs, breaking it down into three distinct categories: recitation (highly duplicated sequences), reconstruction (inherently predictable sequences), and recollection (neither duplicated nor predictable). Through extensive experimentation with the Pythia models ranging from 70M to 12B parameters, the authors demonstrate that different types of memorization exhibit distinct patterns and dependencies on factors like sequence duplication, model size, and training time. They validate their taxonomy by showing its effectiveness in predicting memorization likelihood and reveal that recollection grows disproportionately with model size and training time. The work provides valuable insights into how different factors influence memorization depending on the taxonomic category.

**Strengths:**

The paper exhibits several notable strengths that demonstrate its potential value to the field. The proposed taxonomy of memorization provides an intuitive and practical framework for understanding different types of memorization in LLMs. The extensive experimental validation across model scales and training time points offers valuable insights into how memorization behavior evolves. The authors' approach to validating their taxonomy through predictive modeling and dependency analysis shows methodological rigor and provides empirical support for their theoretical framework.

**Weaknesses:**

1) The template detection approach appears oversimplified. For instance, only basic patterns of "repeating" and "incrementing" sequences are considered, potentially missing more complex templates. The duplication counting relies on exact matches without accounting for semantic similarity or near-duplicates (e.g. slightly modified code or text passages).
2) The paper insufficiently compares its taxonomy against existing memorization frameworks. For example, the relationship between these categories and counterfactual memorization, which is mentioned but not analyzed, deserves exploration. The advantages of this taxonomy over other approaches to studying memorization are not quantitatively demonstrated.
3) The exact procedure for computing KL divergence in Fig 3 is unclear, and the methodology for computing perplexity scores used throughout the analysis lacks essential details. The robustness of results to different tokenization choices is not evaluated.

**Questions:**

1) Can you provide empirical justification for this specific cutoff? How sensitive are your results to this choice?
2) Could you include statistical significance tests for the reported trends across model sizes?

---

> ### Author Response · Authors · 2024-11-19
> **Response**
>
> > The template detection approach appears oversimplified. For instance, only basic patterns of "repeating" and "incrementing" sequences are considered, potentially missing more complex templates. The duplication counting relies on exact matches without accounting for semantic similarity or near-duplicates (e.g. slightly modified code or text passages).
>
> We fully agree that more complex templates are of great interest. In fact, it is one of our core motivations; while a majority of non-recitation examples can be specifically described as either repeating or incrementing, as is made clear in our findings, our goal in qualitatively analyzing recollection was to identify possible other more complex templates. See Section 4.3 for our resulting qualitative analysis, which conjectures that most cases of recollections are cases of extrapolating from slight differences in translation or indexing systems in a unique instance of biblical or legal documents. Such instances can be described as a mix of recitation and reconstruction.
>
> One useful finding is a rejection of a hypothesis (based on Dankers et al. (2023)) about recollection: that a significant amount of rare sequences are simply retokenized versions of more common sequences. By quantifying string similarity, we reject the retokenization hypothesis outright (App B) and instead posit the templating idea above.
>
> > The paper insufficiently compares its taxonomy against existing memorization frameworks. For example, the relationship between these categories and counterfactual memorization, which is mentioned but not analyzed, deserves exploration. The advantages of this taxonomy over other approaches to studying memorization are not quantitatively demonstrated.
>
>
> Our goal here is to analyze individual memorized sequences without relying on potentially intractable definitions of memorization, like those used at smaller scales to guarantee the counterfactual memorization definition. Our stratification of the reconstruction set in particular allows us to exclude many “memorized” samples that are not counterfactually memorized from our analysis of recollection.
>
>
> > The exact procedure for computing KL divergence in Fig 3 is unclear, and the methodology for computing perplexity scores used throughout the analysis lacks essential details. The robustness of results to different tokenization choices is not evaluated.
>
> Do you have specific questions about the procedure for computing perplexity? We use the tokenizer that is native to the LM, and it is unclear to us how we might do otherwise. Perplexity is computed directly by the LM itself; although it is the only feature we use that draws directly on the LM’s behavior, we felt it was important to incorporate because it is so strongly predictive of memorization.
>
> > Can you provide empirical justification for this specific cutoff? How sensitive are your results to this choice?
>
> Please see top level response.
>
> > Could you include statistical significance tests for the reported trends across model sizes?
>
> Using a binomial significance test with p<0.01, we find that every difference between models is significant (with all p-values <<10^-10). We will mention this in our next updated revision.

---

> > ### Comment · Reviewer_FK6z · 2024-11-25
> >
> > Thank you for addressing my concerns—most of my questions have been resolved. However, I noticed an issue with the caption for Figure 3, where the layout seems affected by excessive vspace below it. I recommend revising this for better formatting.

---

> > > ### Author Response · Authors · 2024-11-25
> > > **Thank you**
> > >
> > > Thank you for drawing our attention to the figure caption. We will make sure to fix it in the next draft.
> > >
> > > We are glad to have addressed your concerns, and gratified to learn that most of your questions have been resolved. In light of that, will you consider upgrading your score?

---

> > > > ### Comment · Reviewer_FK6z · 2024-11-25
> > > >
> > > > Thank you for your response. I have updated the score from 6 to 8. Please ensure that the vspace for all figure captions in the paper is correct.

---

### Official Review · Reviewer_bXcZ · 2024-11-04

**Soundness:** 2
**Presentation:** 2
**Contribution:** 2
**Rating:** 6
**Confidence:** 3

**Summary:**

This paper introduces a novel taxonomy for memorization in language models, categorizing it into three types: recitation (highly duplicated sequences), reconstruction (predictable templates), and recollection (other memorized sequences). The authors validate their taxonomy through predictive modeling and analysis across model scales and training time, demonstrating how different factors influence each category distinctly. The work provides valuable insights into understanding memorization as a multifaceted phenomenon rather than a uniform behavior.

**Strengths:**

The proposed memorization taxonomy is intuitive and interesting, drawing parallels with human memorization. This taxonomy is particularly valuable as it provides a structured approach to analyzing what has typically been treated as a uniform phenomenon.

The analysis methodology is another strong point, featuring a thorough examination of dependencies between features and memorization categories, supported by effective predictive modeling to validate the taxonomy.

**Weaknesses:**

The main weakness of this paper boils down to two key issues. First, while the idea of categorizing memorization into three types sounds cool, the paper doesn't dig deep enough to tell us why we should care. Sure, they show that code gets memorized more than text across all categories - but why? And what does this mean for how these models actually work? How different types of memorization contribute to model capabilities. These are the kind of insights that would make the taxonomy actually useful, but they're missing.

In addition, the experimental setup is not convincing. For example, the experiments are conducted solely on Pythia models without validation of other popular models. And some of the key choices seem pretty arbitrary like picking $k=32$ for their memorization tests or saying "more than 5 duplicates" counts as recitation. Why those numbers? What happens if you change them?

Overall, I think the paper lacks insights and the experiments are not very solid.

**Questions:**

What insights from previous work on memorization mechanisms support or conflict with these findings?

---

> ### Author Response · Authors · 2024-11-19
> **Response**
>
> Thank you for your thoughtful review. We are happy you see the value of such a taxonomy!
>
> > First, while the idea of categorizing memorization into three types sounds cool, the paper doesn't dig deep enough to tell us why we should care. Sure, they show that code gets memorized more than text across all categories - but why? And what does this mean for how these models actually work? How different types of memorization contribute to model capabilities. These are the kind of insights that would make the taxonomy actually useful, but they're missing.
>
> Our taxonomic model is just a starting point for analysis, which might eventually involve the inclusion of specific features of interest for future projects. We do point to several insights provided by our more general setting in section 6.3. First, controlling for other factors, rare tokens actually make recollection more difficult—but has no significant effect for other memorization types, indicating that there are in fact cases where a model might have memorized a rare sequence, but the presence of individual rare tokens presented too much friction. We find that recitation is less supported by rare prompts compared to other types of memorization, likely because the recitation set is mostly composed of extremely long predictable sequences. Recitation is also slightly more likely if there are fewer common tokens in the continuation (based on P75 frequency), suggesting that it is difficult to memorize by conditioning on many common tokens. We also find that over the 5-duplicate threshold, the particular number of duplicates becomes unimportant, in spite of how important it is below that threshold; we posit that it takes a small number of exposures for a sequence to become easily memorized, but once memorized, it is rarely forgotten. We are updating our paper to highlight these speculations, which may be confirmed in future causal/interventional experiments.
>
> > In addition, the experimental setup is not convincing. For example, the experiments are conducted solely on Pythia models without validation of other popular models. And some of the key choices seem pretty arbitrary like picking k=32 for their memorization tests or saying "more than 5 duplicates" counts as recitation. Why those numbers? What happens if you change them?
>
> Both the use of Pythia models and the use of k=32 are decisions driven by the availability of memorization data for these models. Unfortunately, other models have not released public memorization datasets, and no models have released public memorization datasets that use different definitions of memorization.
>
> As for the particular duplicate count threshold, please see top level response (1).
>
> > What insights from previous work on memorization mechanisms support or conflict with these findings?
>
> There are several connections we discuss in the paper. We compare to Dankers et al. (2023), who pointed to rare tokens as promoting counterfactual memorization; we find that this relationship is specific to the recollection category, which is likely to contain most counterfactually memorized examples. We also agree with the conclusion of Tirumala et al. (2022) that larger models memorize more sequences.

---

> > ### Comment · Reviewer_bXcZ · 2024-11-26
> >
> > Thanks for your response. My main concern about implications and insights is addressed, and I raise my score accordingly.

---

### Official Review · Reviewer_5vzG · 2024-11-05

**Soundness:** 3
**Presentation:** 3
**Contribution:** 3
**Rating:** 5
**Confidence:** 3

**Summary:**

The paper proposes a taxonomy for model memorization and classifies model memorization into three categories, namely recitation, reconstruction and recollection. The authors identify several data-related or model-related features and test their correlation with model memorization. To verify the effectiveness of the proposed taxonomy, the authors trained three linear regression models for three categories respectively and found that group the regression models attain better performance than the predictors trained on other features.

**Strengths:**

+ A new taxonomy for understanding and analyzing the model memorization.
+ Interesting findings on the dynamics of memorization during the scaling-up of data and model size.
+ An empirical evaluation of the utility of the taxonomy based on predictability.

**Weaknesses:**

+ A perplexing part of the taxonomy is the classification of repetitive or incremental sequences following a specific pattern. If the sequence duplicates more than five times, how do we know whether it is truly "memorized" or it is reproduced simply because the LLM learns its pattern?
+ Why do we use more than five times duplication as the decision boundary for recitation and non-recitation. How is the hyper-parameter decided? Using a single threshold actually assumes an equality in difficulty for reciting every sequence.

**Questions:**

+ The taxonomy of memorization is purely established based on the property of data, i.e., the number of duplications in the pre-training corpus and the implicit template within the data. However, memorization is also a concept and phenomenon related to model behavior and model behaviour can also be included into the taxonomy as an evidence to classify different types of memorization.

---

> ### Author Response · Authors · 2024-11-19
> **Response**
>
> Thank you for your insightful review.
>
> >If the sequence duplicates more than five times, how do we know whether it is truly "memorized" or it is reproduced simply because the LLM learns its pattern?
>
> To address the first weakness, we highlight that current work on memorization often involves the k-elicitation definition, which does not differentiate counterfactual memorization from reconstruction. Obviously, under some circumstances you want to limit notions of memorization to counterfactual cases, and under such circumstances you might develop a different taxonomy from ours, possibly one that becomes more granular about the types of counterfactual memorization. However, one advantage of our approach is that it is far more efficient than existing methods for evaluating counterfactual memorization, as it only requires one training of the model and does not require backwards passes on the data.
>
> > Why do we use more than five times duplication as the decision boundary for recitation and non-recitation. How is the hyper-parameter decided?
>
> Please see top-level response.
>
> > Using a single threshold actually assumes an equality in difficulty for reciting every sequence.
>
> One of our goals in describing the memorization behavior with a taxonomy is to disentangle the factors in “difficulty” of memorizing different types of sequences. By controlling for duplication, we show that factors like rare tokens have specific effects on the likelihood of recitation. (See fig 6)

---

### Author Response · Authors · 2024-11-19
**Top level response regarding threshold hyperparameter**

We thank all reviewers for their insightful reviews, and are pleased that several have mentioned the benefits of an intuitive taxonomy, and see its practical value. We are particularly gratified to see several reviewers recognize our empirical approach as “thorough” (bXcZ) and “methodologically sound” (p6LS) with “methodological rigor” (FK6z).

Several reviewers brought up the choice of duplication threshold for recitation, how we chose that hyperparameter, and how sensitive our model is to its selection. Our collective response to this common criticism is below, and we are revising the paper to more clearly explain how we selected this threshold intuitively.

Please see Fig 3. It illustrates that at around 5-6 duplicates in the corpus, there is a change in the effect of duplication on the difference between memorized and non-memorized perplexity distributions. To walk you through the reasoning, perplexity is the strongest single predictor of memorization (as shown in our work and previous papers). Therefore, stratifying out highly duplicated vs rare samples is justified if it helps to infer memorization, especially from perplexity. In general, a categorical variable for a continuous value is most justified if an important correlation is reversed between those categories.

The fact that 5-6 marks the peak in Fig 3 makes this case clear: for samples below that threshold, higher duplication is associated with greater divergence between memorized and unmemorized perplexities. Above that threshold, higher duplication is associated with lower divergence. We then empirically validate that this threshold is better than a quartile-based cutoff when we compare our memorization predictor to one based on optimal quartile partitions (fig 5).

You do bring up an important point: We don’t compare our threshold to other similarly small ones, only to quartile thresholds. The advantage of our threshold is that it is selected based on the intuitions above, rather than on tuning with an additional variable, but if a different threshold were significantly better, we would have to reassess our judgment on this. To that end, we have run an **additional experiment** revealing that different thresholds for recitation behave similarly to threshold 5, though with minor differences on specific metrics. Setting a recitation threshold at >1 in particular hurts AUC for all categories, and our intuitive threshold of 5 performs best on the aggregate dataset. We have added this figure to our new appendix G.

---

### Meta-Review · Area_Chair_MNeH · 2024-12-21

**Metareview:**

This paper presents a taxonomy for memorization in language models, categorizing it into recitation, reconstruction, and recollection. The authors validate this taxonomy by studying how different factors influence memorization within each category.

All reviewers appreciate the intuitiveness of the taxonomy and soundness of the analysis methodology. The analysis reveals interesting insights such as the disproportionate growth of recollection with model size and training time, and the limited impact of extreme duplication on recitation rates beyond a certain threshold.

The reviewers argue that paper could benefit from a more in-depth discussion of the implications of the taxonomy, particularly how it relates to existing work on memorization mechanisms and how it can inform future research. The experimental setup could be further strengthened by considering a wider range of models and exploring the sensitivity of results to hyperparameter choices.

Despite the said limitations, this paper makes a significant contribution to our understanding of memorization in LLMs and I recommend to accept it as a poster.

**Additional Comments On Reviewer Discussion:**

The reviewers raised concerns related to the importance of even having this taxonomy, and insufficient comparison/establishing context as it relates to prior work. They have mentioned concerns about the simplistic nature of some of the analysis, and lack of insights for memorization of "rare" items. The authors have addressed the majority of the reviewers' concerns effectively and promised to revise the final draft.

---

### Decision · Program_Chairs · 2025-01-22

Accept (Poster)